# Activated Gab1 drives hepatocyte proliferation and anti-apoptosis in liver fibrosis via potential involvement of the HGF/ c-Met signaling axis

Da-eun Nam[1‡], Soo-Jeung Park[1‡], Samson Omole[2], Eugene Um[1], Ramin M. Hakami[2], Young S. Hahn[1,3]*

1 Beirne B. Carter Center for Immunology Research, University of Virginia, Charlottesville, Virginia, United States of America, 2 School of Systems Biology, and Center for Infectious Disease Research, George Mason University, Manassas, Virginia, United States of America, 3 Department of Microbiology, Immunology, and Cancer Biology, University of Virginia, Charlottesville, Virginia, United States of America

‡ DN and SJP are contributed equally to this work as co-first authors.
* ysh5e@virginia.edu

**Data Availability Statement:** All relevant data are within the manuscript and its Supporting information files.

## Abstract

Chronic liver diseases are caused by hepatic viral infection, chemicals, and metabolic stress. The protein Grb2-associated binder 1 (Gab1) binds to various growth factor receptors, and triggers cell differentiation/survival signaling pathways. To identify signaling molecules involved in the progression of liver diseases, we performed reverse-phase protein microarray (RPMA)-based screening of hepatocytes isolated from humanized mice after acute HCV infection. Acute viral infection in humanized liver mice significantly decreased the level of hepatocyte p-Gab1. Moreover, hepatoma cells upon HCV infection decreased Gab1 mRNA at later times of infection (D3 to D5) and p-Gab1 level was inversely related to the production of TGF-β. In contrast, the level of p-Gab1 was increased in CCL4-induced fibrotic liver. Hepatoma cells showed elevation of p-Gab1, along with an increase in STAT3 and ERK activation, upon treatment with HGF (ligand of HGF receptor/c-Met) and CCL4. In Gab1 knockdown hepatoma cells, cell proliferative signaling activity was reduced but the level of activated caspase-3 was increased. These findings suggest that hepatocyte Gab1 expression may play a role in promoting liver fibrosis progression by triggering ERK activation and inhibiting apoptosis. It implies that the Gab1-mediated signaling pathway would be a promising therapeutic target to treat chronic liver diseases.

## Introduction

Hepatocellular carcinoma (HCC) is the most common in cancer-related mortality, and is caused by chronic liver diseases such as liver fibrosis and cirrhosis [1]. Liver fibrosis is developed by hepatocyte damage as a result of various factors such as chronic HCV infection, fatty diet, and toxic chemicals. Apoptosis plays a critical role in removing damaged hepatocytes and

**Funding:** 1. - Y.S.H. - Grant R01 DK122737 - National Institutes of Health This funder participated in overall designing the study, managing research funds, setting the direction of the research, assessing the value of the results, making visual evaluation decisions, and writing and reviewing the mansucript. 2. - R.M.H - Grant R42 AI122666 - National Institutes of Health This funder participated in directing RPMA study, managing research funds, assessing the value of the all data, and providing comments on the manuscript.

**Competing interests:** The authors have declared that no competing interests exist.

maintaining liver homeostasis. Moreover, the increased hepatocyte apoptosis is related to the severity of liver disease. Uncontrolled apoptotic hepatocytes promote liver fibrosis and HCC. In many HCC patients, diagnosis is delayed due to asymptomatic disease in early stages, and those diagnosed at an advanced stage have limited treatment options available to them. An understanding of the molecular mechanisms underlying hepatocyte apoptosis and survival helps the development of therapeutic approaches to prevent the development of serious chronic liver diseases.

Grb2-associated binder 1 (Gab1) is an adapter protein for various growth factors and cytokine receptors including hepatocyte growth factor (HGF), and receptor tyrosine kinase c-Met (c-Met). Gab1 has multi-substrate docking sites and c-Met is one of the well-known substrates. Gab1 contains several potential tyrosine phosphorylation sites and is phosphory-lated in response to various growth factors and cytokines stimulation. Subsequently, phos-phorylated Gab1 (pGab1) can activate mitogenic signaling pathways such as the extracellular signal-regulated kinase (ERK) pathway and the phosphoinositide 3-kinase (PI3K)/AKT pathway [2]. Gab1 has an important role in hepatocyte proliferation in fibrotic liver through promoting tyrosine phosphorylation and regulation of hepatocyte necrosis in mouse models [3–5]. However, a contradictory role of Gab1 has been reported for exerting protective effect in liver fibrosis [6]. Thus, it is important to clarify the role of Gab1 in the progression of liver fibrosis.

Transforming growth factor-beta (TGF-β) is a pleiotropic cytokine and exerts multiple functions during the progression of severe liver diseases. Under normal conditions, TGF-β inhibits cell proliferation and enhances apoptosis while exerting tumor-suppressing activities by inducing cell cycle arrest in early-stage liver disease. TGF-β has a strong carcinogenic effect in late stage of liver disease, contributing to cell migration and invasion and promoting pro-tumorigenic inflammatory responses [7–9]. During chronic liver injury, damaged hepatocytes trigger the activation of hepatic stellate cells (HSCs) [10]. Activated HSCs release large amounts of TGF-β, resulting in the excessive accumulation of extracellular matrix (ECM) components and interference of hepatocytes repair [11].

Moreover, hepatocyte growth factor (HGF) and its receptor, mesenchymal epithelial metastasis factor (c-Met) play a role in various tumor progression and metastasis [12]. HGF is a potent mitogen for primary hepatocytes. Moreover, abnormal c-Met activation promotes the proliferation and metastasis of HCC through various pathways [12, 13]. The high level of c-Met expression affects mortality in HCC patients, and the HGF/c-Met axis is considered to be a prognostic biomarker in HCC patients [14, 15]. Notably, the level of HGF is increased in plasmin from rat hepatocytes, resulting in reduced TGF-β production and subsequent inhibition of hepatic stellate cells (HSC) activation during hepatocyte-HSC co-culture [16]. These studies indicate that crosstalk between the TGF-β and c-Met signaling pathways plays a pathological role during chronic liver diseases to HCC progression.

In this report, we demonstrate that hepatocyte Gab1 expression is regulated during dif-ferent stages of liver disease progression and Gab1 expression is inversely related to TGF-β expression. Notably, Gab1 level is decreased in the early stage of liver disease with the increased level of TGF-β, while opposite results were observed in the liver fibrosis model. Moreover, siGab1 transfection significantly induced apoptosis in damaged hepatocytes, and markedly reduced cell proliferative signaling compared to the control condition. These results indicate that hepatocyte Gab1 expression levels play a role in controlling the severity of HCV-associated chronic liver diseases by regulating the HGF-c-Met signaling axis.

## Materials and methods

### Cell culture and preparations of HCV virus stock

Huh7.5.1 cells, the human hepatoma cell line, are derivative of a permanent cell line, Huh7, established from male hepatoma tissue. Huh7 cell line was derived from liver tissue of a 57-year-old Japanese male and was differentiated into human hepatocellular carcinoma cell line. The liver tissues were obtained with informed consent from a patient. Huh7.5.1 cells were cultured according to the culture method used in our previous study [17]. Huh7.5.1 cells were infected with HCV (JFH-1 strain, genotype 2a) at a multiplicity of infection (MOI) of 0.5. After 5 days of infection, the supernatant was collected and spun down at 1500 rpm for 5 min to pellet dead cells and debris.

The supernatant was collected and frozen at -80°C in cryotubes at 1mL/vial. Huh7.5.1 cells were seeded at a density of $5 \times 10^5$ in 6-well plate and stabilized up to 16 hrs. Huh7.5.1 cells were infected with HCV at 0.5 MOI. After 5 days, cell lysates were prepared and used for analysis.

The mouse hepatoma cell line, Hepa1-6, was obtained from the ATCC. Hepa1-6 cells were cultured as same condition with Huh7.5.1 cells according to the culture method used in our previous study [17]. Cells were treated with carbon tetrachloride (CCL4) and mouse hepatocytes growth factor (mHGF) for 48 hrs and 30 min, respectively. Cell lysates and supernatants were collected and analyzed.

### Mice and HCV inoculation

Human hepatocytes repopulated fah(-/-) rag1(-/-) IL2rgnull on NOD (FRGN) mice were provided by Yecuris (Tualatin, OR, USA). These humanized mice have over 70% human hepatocytes repopulation. 4–5 months old male humanized liver mice were group housed in cages of up to 3 and maintained under a 12 h light/dark cycle with ad libitum access to food and water. The mice were initially maintained on 2-(2-nitro-4-trifluoro-methylbenzoyl)-1,3-cyclohexedione (NTBC) for 5 days, and NBTC was then removed during the infection experiment (7 days), according to the Yecuris protocol. FRGN mice were injected into the tail vein with $1 \times 10^4$ and $1 \times 10^5$ FFU of HCV in 100 μL PBS (3 mice/group). Mice liver tissues were harvested 7 days after virus inoculation. The mice were anesthetized by intraperitoneal injection of avertin (250 mg/kg, 2,2,2-Tribromoethanol, Sigma-Aldrich, St Louis, MO, USA). The animal used in this study were maintained at the animal facility of the University of Virginia School of Medicine. The animal study was approved by the Institutional Animal Care of the University of Virginia School of Medicine. All methods were carried out in accordance with the animal care guidelines and regulations and were approved by the Ethics Committee of the University of Virginia School of Medicine.

### Isolation of humanized liver mice hepatocytes

Harvested liver tissue was collected on ice and washed with HBSS to remove blood. Liver tissue was minced with scalpels or scissors until it was a slurry form (<3 mm). Minced liver tissue were transferred to 50 mL tube containing pre-warmed EGTA buffer (HBSS, 0.5 mM EGTA, 0.5% BSA) and were agitated (100 rpm) in a water bath with shaking for 10 min, 37°C. Tissue was then washed 3 times in HBSS and then placed in pre-warmed digestion buffer (HBSS, 0.05% collagenase IV, 0.5% BSA, 10 mM CaCl2) and agitated (100 rpm) in a water bath with shaking for 30 min, 37°C. Digested tissue was passed through the metal strainer and supernatant collected and filtered through 100 μm cell strainer. Cell suspensions were pooled to 50 mL tube and centrifuged (80 g for 5min, 4°C) and supernatant discarded.

After suspending in culture medium (Williams E with supplements), cells were counted, and viability was assessed using the trypan blue assay. For purification of hepatocytes, 5 mL of cell suspension was gently mounted over a cushion of 15 mL percoll-suspension and centrifuged ($1470 \times g$ for 20min, 4˚C). After aspiration of the supernatant, cell pellets were used for further analysis.

## Reverse-phase protein microarray (RPMA)

We used the RPMA platform developed at George Mason University that has been highly validated in various peer-reviewed publications and successfully applied for various studies, including interrogation of signaling events in response to infection [18]. For RPMA assays, primary hepatocytes were isolated from mice, washed with PBS, and then lysed in a mixture of 25 mL of 2x Novex Tris-Glycine Sample loading buffer SDS (Thermo Fisher Scientific, Waltham, MA, USA), 24 mL of T-PER Tissue protein extraction reagent (Thermo Fisher Scientific), 200 µL of 0.5 M EDTA, 1.3 mL of 1 M DTT, 1X Protease inhibitor cocktail, 1X Phosphatase inhibitor cocktail 1 and 2 (Sigma-Aldrich).

Lysates were kept at -80˚C until they were immobilized onto nitrocellulose coated slides (Grace Bio-labs, Bend, OR) using an Aushon 2470 arrayer (Aushon BioSystems, Billerica, MA). Each sample was printed in technical triplicates along with reference standards used for internal quality control/assurance. To estimate the amount of protein in each sample, selected arrays (one in every 15) were stained with Sypro Ruby Protein Blot Stain (Molecular Probes, Eugene, OR) following manufacturing instructions [19]. Samples for RPMA analyses exhibited mean protein yields of $1.1 \pm 0.24$ µg/mm$^2$ tissue area collected. Prior to antibody staining, the arrays were treated with Reblot Antibody Stripping solution (Chemicon, Temecula, CA) for 15 min at room temperature, washed with PBS and incubated for four hours in I-block (Tropix, Bedford, MA). Arrays were then probed with 3% hydrogen peroxide, a biotin blocking system (Dako Cytomation, Carpinteria, CA), and an additional serum free protein block (Dako Cytomation) using an automated system (Dako Cytomation) as previously described [18]. Each array was then probed with one antibody targeting an unmodified or a post-translationally modified epitope. Antibodies were validated as previously described [20]. Slides were then probed with a biotinylated secondary antibody matching the species of the primary antibody (anti-rabbit and anti-human, Vector Laboratories, Inc. Burlingame, CA; anti-mouse, CSA; Dako Cytomation). A commercially available tyramide-based avidin/biotin amplification kit (CSA; Dako Cytomation) coupled with the IRDye680RD Streptavidin fluorescent dye (LI-COR Biosciences, Lincoln, NE) was employed to amplify the detection of the signal. Slides were scanned on a laser scanner (TECAN, Mönnedorf, Switzerland) using the 620 nm and 580 nm wavelength channels for antibodies and total protein slides, respectively. Images were analyzed with a commercially available software (MicroVigene 5.1.0.0; Vigenetech, Carlisle, MA) as previously described [18]; this software performs automatic spot finding and subtraction of the local background along with the unspecific binding generated by the secondary antibody. Finally, each sample was normalized to its corresponding amount of protein derived from the Sypro Ruby stained slides and technical replicates were averaged.

A total of 27 samples were analyzed, with three samples collected from each mouse. The samples were collected from two mice treated with PBS, four mice treated with HCV $1 \times 10^4$ FFU, and three mice treated with HCV $1 \times 10^5$ FFU. There were 150 antibodies used in the Reverse-phase protein microarray data study (Table 1). The expression values for each sample were normalized by calculating fold change values relative to the PBS group.

**Table 1. Human-specific antibody sets for reverse-phase protein microarray (RPMA).**

| | | | | | | | | | |
|---|---|---|---|---|---|---|---|---|---|
| 1 | 4EBP1 S65 | 31 | CD3 epsilon | 61 | H2A.X S139 | 91 | MSH2 | 121 | Progesterone Rec S190 |
| 2 | 4EBP1 T70 | 32 | Chk-1 S345 | 62 | HER2 total | 92 | MSK1 S360 | 122 | PTEN S380 |
| 3 | Acetyl-CoA Carboxylase S79 | 33 | Cofilin S3 | 63 | HER2 Y1248 | 93 | mTOR S2448 | 123 | PTEN total |
| 4 | AKT S473 | 34 | CREB S133 | 64 | HER2 Y877 | 94 | NF-kB p65 S536 | 124 | Raf S259 |
| 5 | AKT T308 | 35 | Cyclin A | 65 | HER3 total | 95 | Nrf2 | 125 | Ras-GRF1 S916 |
| 6 | ALK Y1586 | 36 | Cyclin B1 | 66 | HER3 Y1289 | 96 | p27 T187 | 126 | Rb S780 |
| 7 | ALK Y1604 | 37 | Cyclin D1 | 67 | HER4 total | 97 | p38 MAPK T180/Y182 | 127 | Ret Y905 |
| 8 | AMPKa1 S485 | 38 | EGFR total | 68 | HER4 Y1284 | 98 | p53 S15 | 128 | Ron Y1353 |
| 9 | Androgen Rec S650 | 39 | EGFR Y1068 | 69 | Heregulin | 99 | p53 total | 129 | RSK3 T356/S360 |
| 10 | Androgen Rec S81 | 40 | EGFR Y1148 | 70 | HIF-1a | 100 | p70S6K S371 | 130 | S6 Ribosomal Prot S235/36 |
| 11 | ARV-7 | 41 | EGFR Y1173 | 71 | Histone H3 S10 | 101 | p70S6K T389 | 131 | S6 Ribosomal Prot S240/44 |
| 12 | ASK1 S83 | 42 | EGFR Y992 | 72 | HLA-DR | 102 | p70S6K T412 | 132 | SAPK/JNK T183/Y185 |
| 13 | ATM S1981 | 43 | eIF4E S209 | 73 | HLA-DR/DP/DQ/DX | 103 | p90RSK S380 | 133 | SGK1 S78 |
| 14 | ATP-Citrate Lyase S454 | 44 | eIF4G S1108 | 74 | HSP90a T5/7 | 104 | p90RSK T359/S363 | 134 | Shc Y317 |
| 15 | ATR S428 | 45 | Elk-1 S383 | 75 | IGF1R/Insulin Rec Y1131/Y1146 | 105 | PAK1/2 S199/204 S192/197 | 135 | pSmad2 S245/250/255 |
| 16 | Aurora A/B/C T288/232/198 | 46 | eNOS S113 | 76 | IGF1R/Insulin Rec Y1135/36 Y1150/51 | 106 | PAK1/2 T423/402 | 136 | Src Family Y716 |
| 17 | B-Raf S445 | 47 | eNOS S1177 | 77 | IkB-a S32/36 | 107 | PARP, cleaved D214 | 137 | Src Y527 |
| 18 | BAD S112 | 48 | eNOS/NOS III S116 | 78 | Insulin Rec B | 108 | PD-L1 | 138 | Stat1 Y701 |
| 19 | Bcl-2 S70 | 49 | ERK1/2 T202/Y204 | 79 | IRS-1 S612 | 109 | PDGFRa Y754 | 139 | Stat3 S727 |
| 20 | BRCA1/2 S1524 | 50 | Estrogen Rec a S118 | 80 | IRS-1 total | 110 | PDGFRb Y716 | 140 | Stat3 Y705 |
| 21 | c-Abl T735 | 51 | Estrogen Rec a Total | 81 | Jak1 Y1022/23 | 111 | PDGFRb Y751 | 141 | Stat4 Y693 |
| 22 | c-Abl Y245 | 52 | FADD S194 | 82 | Jak2 Y1007 | 112 | PDK1 S241 | 142 | Stat5 Y694 |
| 23 | c-Kit Y703 | 53 | FAK Y576/577 | 83 | Ki67 (MIB-1) | 113 | PI3K p85/55 Y458/199 | 143 | Stat6 Y641 |
| 24 | c-Kit Y719 | 54 | FGF Rec Y653/654 | 84 | LC3B | 114 | PKA C T197 | 144 | TROP2 |
| 25 | c-PLA2 S505 | 55 | FOXM1 T600 | 85 | LKB1 S334 | 115 | PKC a S657 | 145 | Tuberin/TSC2 Y1571 |
| 26 | c-Raf S338 | 56 | FOXO1 S256 | 86 | M-CSF Rec Y723 | 116 | PLCgamma1 Y783 | 146 | Tyk2 Y1054/1055 |
| 27 | Caspase-3, cleaved D175 | 57 | FOXO1/O3a T24/32 | 87 | MDM2 S166 | 117 | PP2A a Subunit | 147 | VEGFR2 Y1175 |
| 28 | Caspase-9, cleaved D330 | 58 | FOXO3a S253 | 88 | MEK1/2 S217/221 | 118 | PP2A B Subunit | 148 | VEGFR2 Y951 |
| 29 | Catenin B S33/37/T41 | 59 | Gab1 Y627 | 89 | pMet Y1234/35 | 119 | PRAS40 T246 | 149 | VEGFR2 Y996 |
| 30 | Catenin B T41/S45 | 60 | GSK3aB S21/9 | 90 | MLH1 | 120 | PRK1/2 T774/816 | 150 | YAP S127 |

## Carbon tetrachloride (CCL4)-induced liver fibrosis mouse model

Male (4–6 wk) C57BL/6J mice were purchased from Jackson Laboratory (Bar Harbor, ME, USA). Mice were group housed in cages of up to 3 and maintained under a 12 h light/dark cycle with ad libitum access to food and water. Mice were administered with CCL4 every 3 days via oral gavage, receiving escalating doses over time; 0.875 mL/kg (1st dose, week 1), 1.75 mL/kg (2nd to 9th dose, week 1–4), and 2.5 mL/kg (10th to 23rd dose, week 4–6). Liver tissues were frozen immediately in liquid nitrogen, and were either fixed in 10% formalin or embedded in Tissue-Tek OCT compound (Sakura Finetek, Torrance, CA, USA) for immunohistochemistry analysis.

## Histological study

Collected tissues were fixed in 10% buffered formalin for 24 hours. Serial 5-μm paraffin sections were prepared and either stained with hematoxylin and eosin (H&E) or picrosirius red staining. To perform picrosirius red staining, dewax formalin fixed paraffin-embedded tissue sections were stained in Sirius red working solution for 1hour, and were washed. Slides were dehydrated by immersing them through the alcohol solution. After immersing slides in xylene solution, slides were mounted by using ProLongTM Gold antifade reagent (Thermo Fisher Scientific) and cover slips. The sections were analyzed using an Olympus BX40F4 microscope (Southern Microscope, Inc., Haw River, NC, USA).

## RNA extraction and quantitative real-time reverse transcription polymerase chain reaction (RT-qPCR) analysis

Huh7.5.1 cells, Hepa1-6 cells, and isolated mouse hepatocytes were lysed in the presence of Trizol. Total RNA was extracted from the tissues or primary hepatocytes using a miRNeasy mini kit, according to the manufacturer's protocols (Qiagen, Valencia, CA, USA). RNA concentration and quality were determined using a Nanodrop 2000 (Thermo Fisher Scientific). Only samples showing a 260/280 nm ratio between 1.8 and 2.1 were selected for cDNA transcription. A total of 1μg of RNA, and a total of 10 ng of RNA (for miRNA assay) was used for cDNA synthesis. Complementary DNA (cDNA) was then synthesized using High-Capacity cDNA Reverse Transcription Kits (Thermo Fisher Scientific) for mRNA assay. Real-time PCR (Applied Biosystems, Foster City, CA, USA) was performed on triplicate samples using 1 μL of cDNA with Fast SYBR Green Master Mix (Thermo Fisher Scientific) for mRNA assay, or 1.33 μL with the TaqMan® Universal PCR Master Mix, no AmpErase® UNG (Thermo Fisher Scientific). The cDNA for mRNA assay was amplified for 40 cycles of denaturation (95°C for 3 s) and annealing and extension (60°C for 30 s) using the primers shown in Table 2. The

**Table 2. Human and mouse-specific primer sets for RT-qPCR.**

| Name | Forward (5'→3') | Reverse (5'→3') |
|---|---|---|
| HPRT (H) | CCTGGCGTCGTGATTAGTGA | CGAGCAAGACGTTCAGTCCT |
| mTOR (H) | CTTAGAGGACACCGGGGAAG | TCCAAGCATCTTGCCCTGAG |
| STAT3 (H) | CTGTGGGAAGAATCACGCCT | ACATCCTGAAGGTGCTGCTC |
| cRAF(RAF1) (H) | CCTGGCTCCCTCAGGTTTAAG | GAGCCATCAAACACGGCATC |
| Met (H) | TGGCATGTCAACATCGCTCT | AGGAATGCAGGAATCCCACC |
| Gab1 (H) | CTGCCATTAACTGTGCTTCCC | GCTGGCTGGAGGAGTAACAG |
| ERK2 (H) | CCCTGCCTGTGTGCACTTAT | TGGTCCGTAGCCAGTTGTTC |
| TGF-b (H) | GACTTCAGCCTGGACAACGA | TGTAGGGGTAGGAGAAGCCC |
| HPRT (M) | GTTGGGCTTACCTCACTGCT | TAATCACGACGCTGGGACTG |
| mGab1 (M) | GAAGCTTGGACGGATGGGAG | TCCACTCAGATCTCGTCTTCCC |
| TIMP1 (M) | GATCGGGGGCTCCTAGAGACA | GCTGGTATAAGGTGGTCTCGT |
| COL1a1 (M) | TTCTCCTGGCAAAGACGGAC | CCATCGGTCATGCTCTCTCC |
| COL1a2 (M) | CTTGCTGGCCTACATGGTGA | ATGAGTTCTTCGCTGGGGTG |
| Met (M) | GGGAACTGGCTACTGCTCTG | CGTGAAGTTGGGGAGCTGAT |
| mTOR (M) | CGCTCACTGCTGTGCTCTAT | GTAGCGGATATCAGGGTCAGG |
| TGFb1 (M) | ACTGGAGTTGTACGGCAGTG | GGGGCTGATCCCGTTGATTT |
| STAT3 (M) | TGTGTGACACCATTCATTGATGC | GGGAAAGGAAGGCAGGTTGA |
| mRaf1 (M) | GGATAGCCTGAGAGCGTCTTC | AAGAATCCGTGAGCTTGCCA |
| MAPK1 (ERK2) (M) | AATTGGTCAGGACAAGGGCTC | GAGTGGGTAAGCTGAGACGG |
| MAPK3 (ERK1) (M) | ACACTGGCTTTCTGACGGAG | TGATGCGCTTGTTTGGGTTG |

reaction was performed using StepOne Real-Time PCR Systems (Applied Biosystems). HPRT was used as an endogenous control for normalization in Huh7.5.1 cells and primary hepatocytes.

## Western blotting

Huh7.5.1 cells, Hepa1-6 cells, and isolated mouse hepatocytes were lysed in a RIPA Lysis Buffer System (Santa Cruz Biotechnology, Santa Cruz, CA, USA), and centrifuged at $12,000 \times g$ for 20 min at 4°C. The protein content of the clear lysates was measured by using a Pierce BCA Protein Assay Kit (Thermo Fisher Scientific). 60 μg of total protein was added to NuPAGE® LDS sample buffer (4×; Life Technologies, Carlsbad, CA, USA), and the Protein samples were loaded on a 10% SDS-polyacrylamide gel (Bio-Rad Laboratories, Hercules, CA, USA) and then transferred to a nitrocellulose membrane (Bio-Rad Laboratories). Membranes were incubated for 1 hr in a blocking solution, and then incubated for 12 hrs at 4°C with antibodies recognizing either Ki67, CYP2E1 (Abcam, Cambridge, MA, USA), Gab1, pGab1, pSTAT3, pERK1/2, pro-caspase-3, cleaved caspase-3, or GAPDH (glyceraldehyde-3-phosphate dehydrogenase) (Cell Signaling Technology, Danvers, MA, USA). After incubation with the primary antibodies, the membranes were exposed with a secondary antibody (anti-rabbit IgG HRP-linked antibody, 1: 5,000, Cell Signaling Technology) for 1 hr at room temperature. The immunoreactive protein bands were then visualized and protein expression was quantified using the Image J software.

## Hydroxyproline assay

Mouse liver tissues weighing 30–50 mg were used to determine their hydroxyproline content according to the protocol for Hydroxyproline assay kit (Abcam, Cambridge, MA, USA).

## Immunofluorescence microscopy

Huh7.5.1 cells were seeded at a density of $1 \times 10^5$ in 2-well Nunc™ Lab-Tek™ chamber slides (Thermo Fisher Scientific) and stabilized up to 16 hrs. Subsequently, the cells were infected with HCV(JFH-1) at 0.5 MOI for 5 days. Hepa1-6 cells were seeded at a density of $1 \times 10^5$ in 2-well glass chamber slide and stabilized up to 16 hrs, and subsequently treated with 20 mM of CCL4 for 48hrs and 20 mg/mL of mHGF for 30 min. Cells were then fixed using 4% paraformaldehyde reagent for 15 min and then washed. They were blocked and permeabilized according to the method used in our previous study [17]. The cells were then incubated for 12 hrs at 4°C with primary antibodies recognizing pGab1 (Cell Signaling Technology) and HCV core (Invitrogen). After incubation with the primary antibody, cells were incubated with a secondary antibody (Alexa Fluor 488, and Alexa Fluor 555; Invitrogen) at a final dilution of 1:1000 for 1 hr at room temperature. The cells were washed, and overlaid with 4′,6-diamidino-2-phenylindole (DAPI) (Sigma-Aldrich). Images were captured using a LSM700 (Carl Zeiss, Oberkochen, Germany) confocal microscope.

## siRNA transfection

Hepa1-6 cells were seeded at a density of $2 \times 10^5$ cells per well in 6-well culture plates. Cells were treated with CCL4 and transfected with either AllStars Negative Control siRNA (5nM) (Qiagen) or Gab1 siRNA (20pM) (Sigma-Aldrich) using Lipofectamine™ 2000 Transfection Reagent (Thermo Fisher Scientific), following the manufacturer's protocol. Mouse hepatocytes growth factor (mHGF) was added to the cells on 2 days post transfection for 30min, and cell lysates were used for analysis.

### Ethics approval

All methods were carried out in accordance with relevant guidelines and regulation and were approved the University of Virginia IRB committee. Experiments are conducted in compliance with the ARRIVE guidelines.

## Results

### Acute hepatic viral infection in humanized liver mice increases hepatocyte TGF-β signaling-related proteins and leads to decreased levels of pGab1

TGF-β plays a pivotal role in the progression of severe liver diseases. To quantify the impact of hepatic viral infection on *in vivo* regulation of cellular signaling, including TGF-β signaling, we performed reverse-phase protein microarray (RPMA)-based screening of hepatocytes isolated from humanized mice with acute HCV infection. These mice are transplanted with human hepatocytes and develop mild hepatic inflammation after infection. At 7 days after viral infection, hepatocytes were isolated and cell lysates were arrayed onto nitrocellulose coated slides for quantitative probing with a total of 150 antibodies specific for cell signaling proteins (Fig 1A). HCV infection of humanized liver mice was verified by Western blot analysis of core protein (S1 Fig). The data analysis was performed as follows; for changes in total protein expression levels, a difference of 2 fold or higher in either direction (up or down) was considered significant. For changes in phosphorylation levels, a difference of 1.5 fold or higher in either direction (up or down) was considered significant.

Infected hepatocytes showed 15 proteins with significant increase in either total protein levels (2 fold or higher) or phosphorylation (1.5-fold or higher), and PP2A a/b, IRS-1, and Gab1 Y627 with significant decrease in either total protein levels or phosphorylation compared to uninfected cells (Fig 1B). Notably, increased levels of the following proteins that are involved in cell survival and proliferation are found in infected hepatocytes; 1) mTOR and its downstream signaling protein P70S6K, 4EBP1, elF4E; 2) ERK, c-Raf (part of the ERK1/2 pathway); 3) Bcl-2, and STAT3. Moreover, infected hepatocytes show an increased level of transcription factor FOXO1, which is involved in increasing the expression of PEPCK and glycogen-6-phosphatase in hepatocytes. However, pGab1 (Y627) in infected hepatocytes was significantly decreased compared to the uninfected control (Fig 1B and 1C). Taken together, RPMA analysis of hepatocytes after acute viral infection in mice with humanized liver showed decreased levels of pGab1 (Y627), together with concomitant increased activation of TGF-β-related cell survival signaling pathways.

We next validated RPMA results by both qPCR and western blot analysis of infected and uninfected hepatocytes isolated from mice with humanized liver. Consistent with the RPMA results, mRNA expression of mTOR, ERK, and c-Raf was increased in infected cells compared to the control. However, there's no statistical difference in mRNA expression of ERK (Fig 2A). Interestingly, TGF-β mRNA expression level was higher in infected cells compared to control (Fig 2B). Caspase-9 and caspase-3 are enzymes that play a crucial role in apoptosis. Caspase-9 initiates the apoptotic pathway and activates caspase-3, which carries out the final stages of cell death. According to the RPMA results, cleaved caspase-3 ($p = 0.31$) and caspase-9 ($p = 0.0001$) were found to have increased in the HCV-infected group compared to the PBS group (Fig 2C).

Taken together, these results suggest that apoptosis and proliferation signals are simultaneously activated in hepatocytes following viral infection. Based on these results, we have formulated a hypothesis that Gab1 may have a role in regulating the severity of liver disease progression by interacting with the TGF-β signaling pathway.

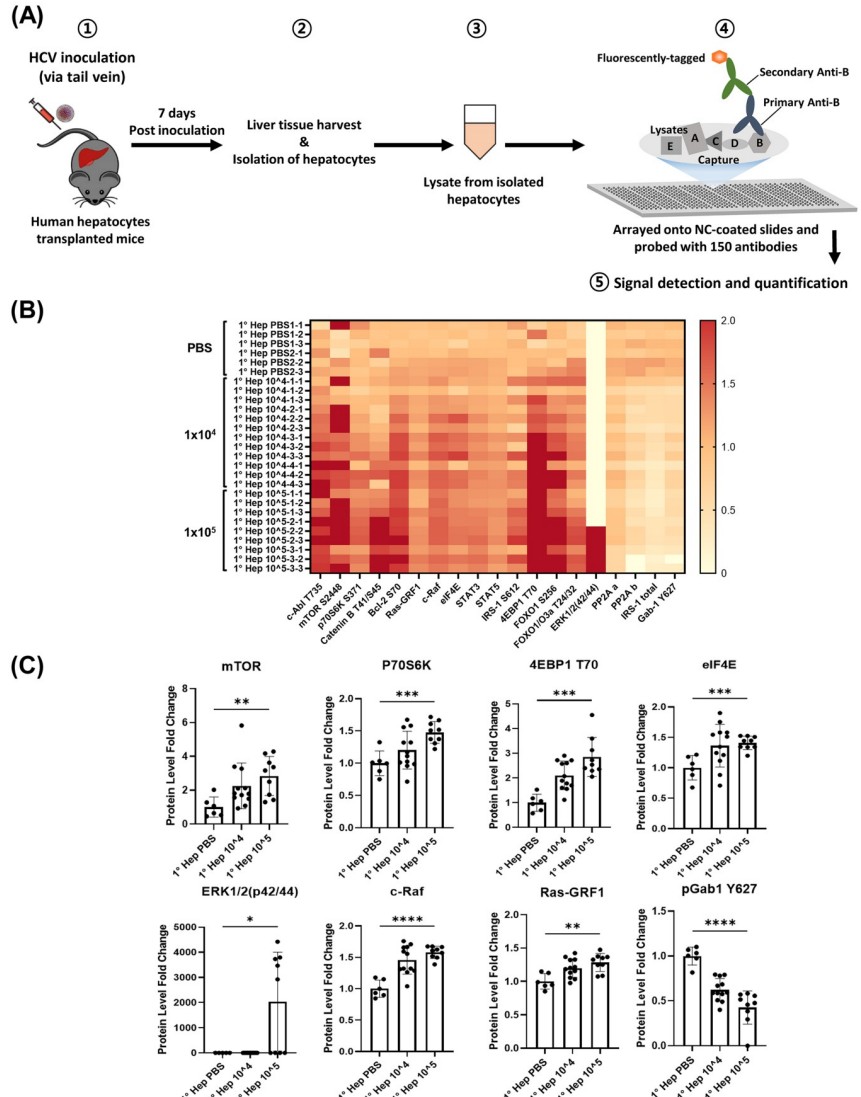

**Fig 1. Early HCV infection enhanced cell survival signaling pathways associated with TGF-β in hepatocytes while phosphorylated Gab1 decreased.** Humanized-liver mice were injected into the tail vein with $1 \times 10^4$ and $1 \times 10^5$ FFU HCV in 100 μL PBS. Liver tissues were harvested 7 days post infection (dpi) and hepatocytes were isolated. Lysates were prepared from isolated hepatocytes and used for reverse-phase protein microarray analysis (RPMA). (A) Schematic diagram showing the RPMA process. (B, C) Heatmap showing the protein fold change in isolated hepatocytes from HCV-infected humanized liver mouse. Data are expressed as mean ± SD (n = 3). *$P < 0.05$, ** $P < 0.01$ and *** $P < 0.001$ were considered statistically significant, as assessed using a t-test.

## pGab1 expression levels are inversely related to TGF-β synthesis during acute viral infection of hepatoma cells

To determine the relationship between Gab1 and TGF-β synthesis, we carried out kinetic studies for comparing the expression of Gab1 with cell proliferation-related signaling molecules as well as TGF-β production at various times of post infection; D0, D1 to D3 (early time points), D4, D5 (late time points). First, we examined Gab1 expression in Huh7.5.1 cells infected with HCV at 0.5 MOI for up to 5 days. Gab1 mRNA increased significantly until D1 but decreased during subsequent days (Fig 3A). The level of pGab1 exhibited a pattern similar to that of

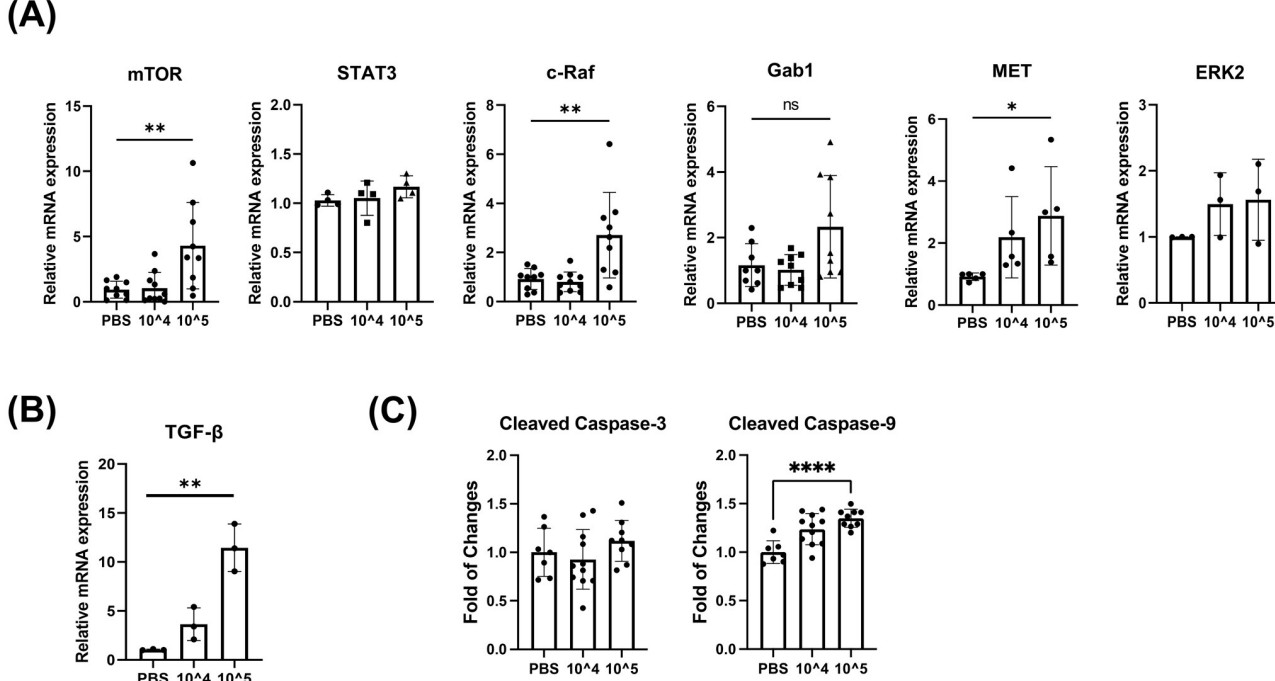

**Fig 2. pGab1 expression levels are inversely related to TGF-β synthesis during acute viral infection of hepatocytes.** Humanized-liver mice were injected into the tail vein with $1 \times 10^4$ and $1 \times 10^5$ FFU of HCV in 100 μL PBS. Liver tissues were harvested 7 dpi and hepatocytes were isolated. (A) mRNA was extracted using TRIzol, and mRNA expression levels were analyzed by qRT-PCR. HPRT was used as an endogenous control for normalization in cells. (B) mRNA expression of TGF-β was analyzed by qRT-PCR. Data are expressed as mean ± SD (n = 3). (C) The expression of caspase-3 ($P = 0.31$) and caspase-9 ($P = 0.0001$) was analyzed by RPMA. *$P < 0.05$, ** $P < 0.01$ and *** $P < 0.001$ were considered statistically significant, as assessed using a t-test.

Gab1 mRNA expression (Fig 3B and 3C). Additionally, the mRNA levels of cell proliferation-related genes, mTOR, STAT3, c-Raf, MET, and ERK showed a similar trend to that of Gab1 (Fig 3D). This was in particular the case for the pSTAT3 protein (Fig 3E).

However, in contrast to Gab1, the mRNA and release protein levels for TGF-β increased over time (Fig 3F). Interestingly, this trend is consistent with our observation of increased TGF-β mRNA and decreased pGab1(Y627) protein levels in HCV-infected hepatocytes as compared to control hepatocytes isolated from humanized liver mice (Fig 2B). These results indicate that Gab1 expression is inversely related to the production of TGF-β during HCV infection in hepatocytes. It suggests that Gab1 activation may play a role in regulation of TGF-β synthesis and production under the condition of acute viral infection in hepatoma cells.

## pGab1(Y627) levels are elevated in the liver of fibrotic mice

Given the role of TGF-β in severe chronic liver disease, and the relation of Gab1 to TGF-β production, we hypothesize that Gab1 may play a role in regulation of liver fibrosis development. To test this, we determined a pathologic role of Gab1 in the progression of liver fibrosis *in vivo* using the established murine model of liver fibrosis induced by CCL4 treatment. Mice were administered with CCL4 orally every 3 days for 6 weeks, then liver tissue was harvested and used for analysis (Fig 4A). As shown in Fig 4B, liver weight and collagen deposition levels were significantly increased in the CCL4 group compared to the control group. Moreover, region of picrosirius red staining was increased in the CCL4 group compared to the control group (Fig

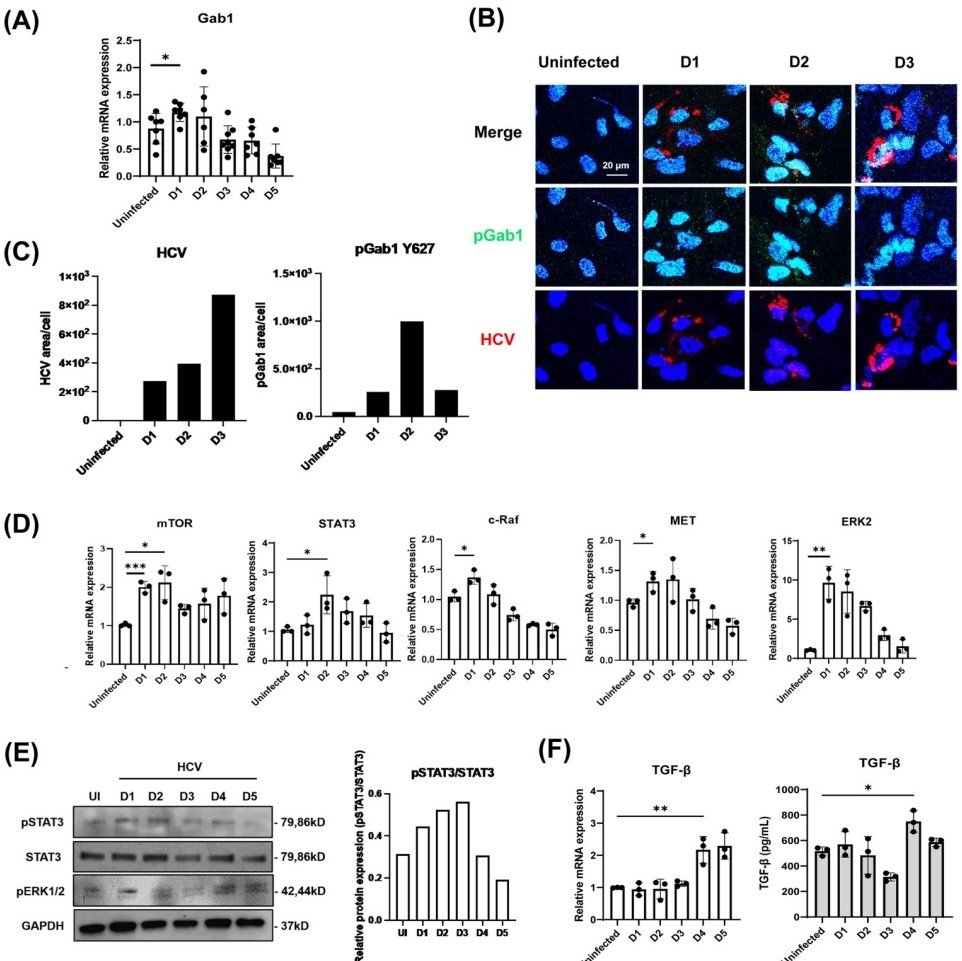

**Fig 3. pGab1 levels decreased over time in HCV-infected Huh7.5.1 cells.** Huh7.5.1 cells were infected with HCV at 0.5 MOI. Cell lysates were collected daily for 5 days for analysis. (A) mRNA was extracted using TRIzol and mRNA expression of Gab1 was analyzed by qRT-PCR. HPRT was used as an endogenous control for normalization in cells. (B) Intracellular pGab1 (Y627) protein levels were assessed by confocal microscopy. Confocal microscopic images of HCV core (red) and pGab1 (Y627) (green) expression in Huh7.5.1 cells for 3 dpi are presented. Nuclei were stained with DAPI. Scale bar, 20 μm. (C) Quantification of the pGab1 and HCV positive areas per Huh7.5.1 cell for either uninfected or HCV-infected cells for D1, D2, and D3. Using ImageJ, the area and the count for pGab1 and HCV core protein in the cells were quantified. (D) mRNA expression of mTOR, STAT3, c-Raf, MET, and ERK2 were measured by qRT-PCR. (E) Protein levels of pSTAT3, STAT3, pERK1/2 and GAPDH were assessed by western blot analysis. (F) mRNA expression of TGF-β was analyzed by qRT-PCR, and TGF-β protein levels were measured by ELISA using cell culture supernatant. Data are expressed as mean ± SD (n = 3). *$P < 0.05$, ** $P < 0.01$ and *** $P < 0.001$ were considered statistically significant, as assessed using a t-test.

4C), indicating the CCL4-induced development of liver fibrosis. Surprisingly, Gab1 mRNA and relative protein levels are significantly increased in CCL4-induced liver fibrosis mice compared to those in control mice (Fig 4D and 4E).

We next determined the expression of signaling molecules downstream of Gab1. Notably, protein levels of pSTAT3, CYP2E1, and pERK1/2 were significantly increased in the CCL4 group as compared to the control group (Fig 4F). Moreover, there was significantly increased TGF-β, mTOR, STAT3, c-Raf mRNA expression, without any differences in ERK1/2 or MET mRNA levels (Fig 4G and 4H). Taken together, these findings suggest that Gab1 may play an important role in progression of chronic liver disease by regulating cell proliferation.

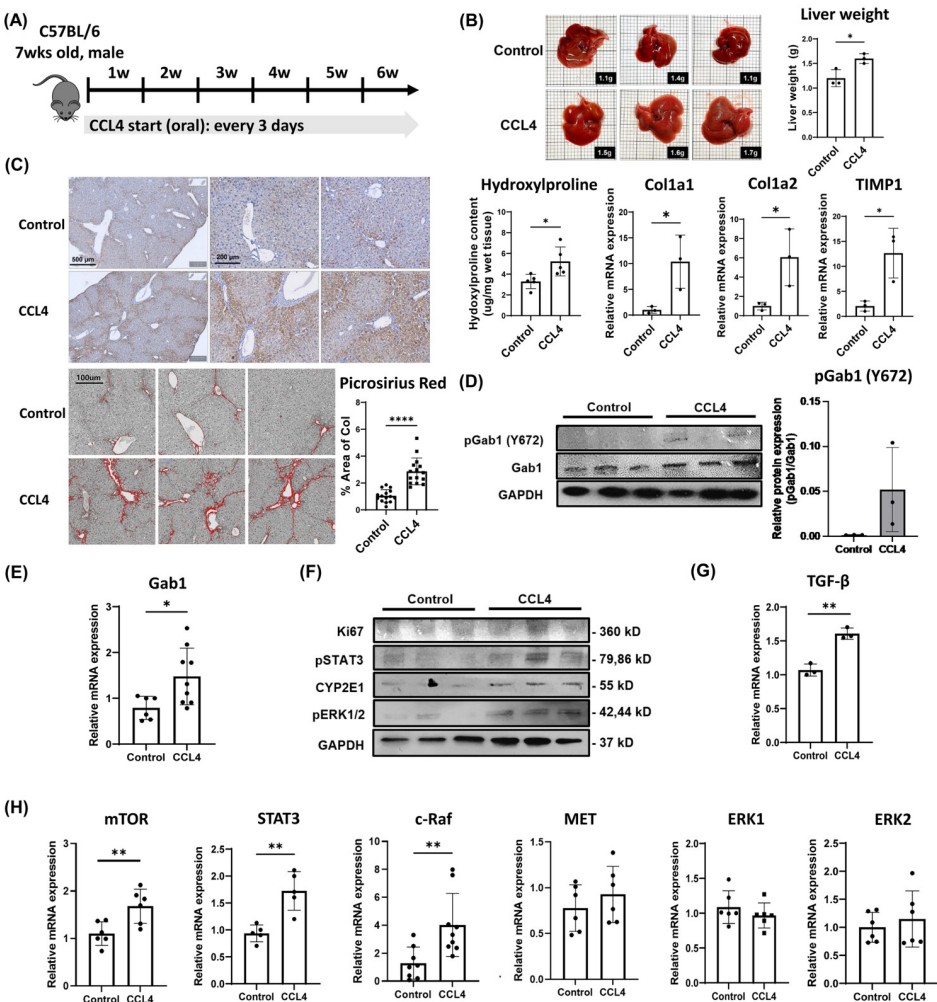

**Fig 4. pGab1(Y627) levels are elevated during the development of liver fibrosis.** (A) Mice were administered with CCL4 every 3 days orally, receiving escalating doses over time. Liver tissues were collected for analysis 6 weeks after CCL4 induction of liver fibrosis. (B) Hydroxyproline level was analyzed using wet liver tissues (30–50 mg). mRNA was extracted using TRIzol, and mRNA expression was analyzed by qRT-PCR. HPRT was used as an endogenous control for normalization in cells. (C) Liver tissue was collected and fixed in 10% buffered formalin for 24 hours. Serial 5-μm paraffin sections were prepared for staining with H&E and picrosirius red. Quantification of red-stained collagen in an image of a mouse liver tissue section stained with Sirius Red using the ImageJ program is shown. To obtain statistical results, we repeated the quantification five times in different areas of liver tissue sections (n = 15). (D and E) mRNA and protein expression levels of Gab1 and pGab1 (Y627) were analyzed by qRT-PCR and western blot respectively. (F) Protein expression of Ki67, pSTAT3, CYP2E1, pERK1/2, and GAPDH was analyzed by western blot. (G) mRNA expression levels of TGF-β were measured by qRT-PCR (n = 3). (H) mRNA expression levels of mTOR, STAT3, c-Raf, MET, and ERK2 were measured by qRT-PCR (n = 6). *P < 0.05, ** P < 0.01 and *** P < 0.001 were considered statistically significant, as assessed using a t-test.

## Gab1 activation promotes hepatocyte proliferation and survival via the HGF-c-MET signaling axis

HGF/c-Met signaling axis plays a role in tumor progression and metastasis. The functional regulation between TGF-β and HGF/c-Met signaling is crucial for HCC progression [16]. Gab1 is an adapter protein that binds c-Met and promotes cell survival and proliferation [21]. Thus, Gab1 may affect HGF/c-Met signaling and trigger cell proliferation during liver fibrosis.

To test this, we determined Gab1 expression following co-treatment with CCL4 and HGF using the mouse hepatoma cell line Hepa1-6 cells. Interestingly, there was a significant increase in Gab1 mRNA and pGab1(Y627) protein levels in CCL4 and HGF co-treated cells compared to the control cells (Fig 5A–5C). The STAT3 mRNA levels was only increased in HGF treated cells while c-Raf mRNA significantly increased in CCL4/HGF treated cells as compared to the control condition (Fig 5D). No effects were observed on mTOR, MET, and ERK1/2 mRNA expression following CCL4 or CCL4+HGF treatments (Fig 5D). These results suggest that HGF and CCL4 may cooperate for the activation of Gab1 protein.

Next, to clarify the functional role of Gab1, we transfected Hepa1-6 cells with Gab1 short interfering RNAs (siGab1) to analyze the effect of decreasing Gab1 expression levels. The siGab1-transfected cells showed significantly decreased Gab1 mRNA and protein expression levels compared to the siCon cells, which were treated with negative control siRNA (Fig 6A and 6B). Gab1 knockdown cells showed an enhanced apoptosis signal, with increased protein levels for cleaved caspase-3, and decreased levels of pSTAT3 and pERK1/2 proteins (Fig 6C). Moreover, TGF-β mRNA level was increased in siGab1 cells compared to siCon cells (Fig 6D). These results suggest that the expression of hepatocyte Gab1 is induced by HGF-c-Met signaling, and is involved in increasing cell proliferation/survival and TGF-β expression.

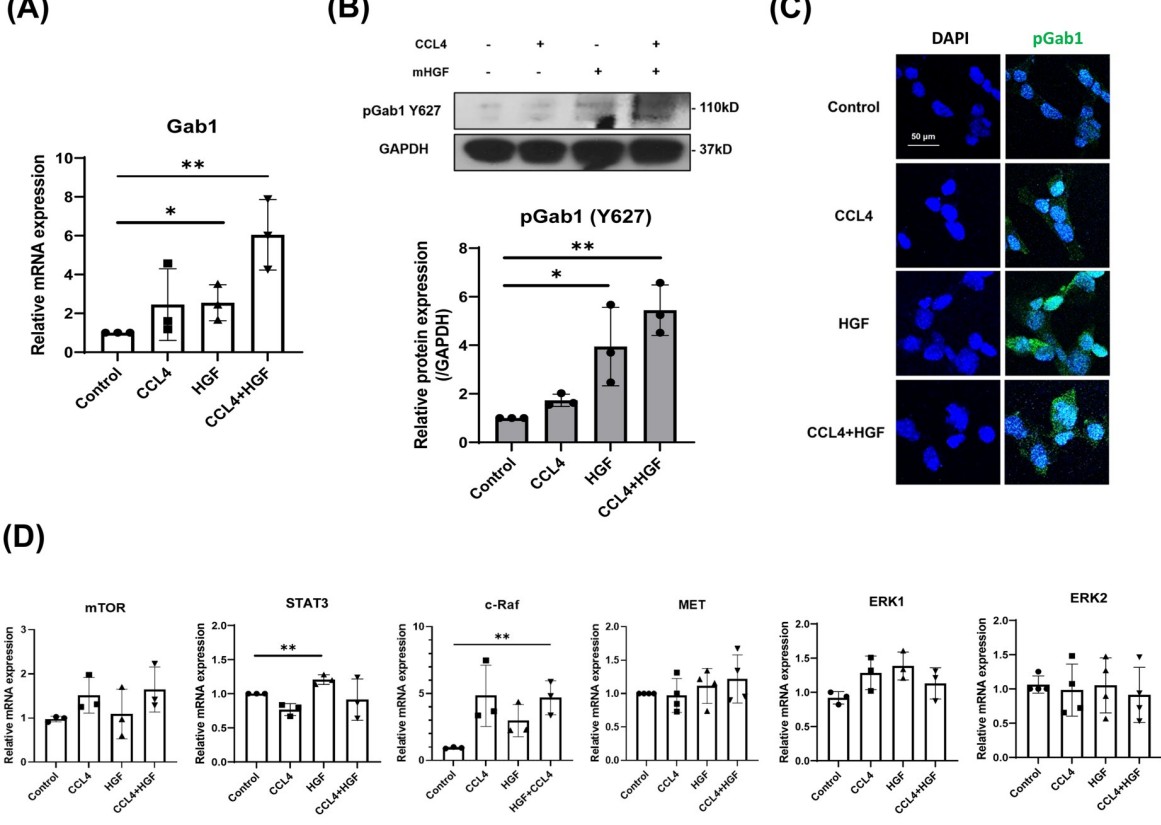

**Fig 5. Gab1 expression increases in mouse hepatoma cell line treated with CCL4 and HGF.** Hepa1-6 cells were treated with 20 mM of CCL4 and 20 mg/mL of mouse hepatocytes growth factor (mHGF) for 48 hrs and 30 min respectively. Cell lysates and culture supernatants were collected and analyzed. (A) mRNA expression of Gab1 was measured by qRT-PCR. (B) Protein levels of pGab1 and GAPDH were analyzed by western blot. (C) Intracellular pGab1 (Y627) protein levels were analyzed by confocal microscopy. Confocal microscopic images of pGab1 (Y627) (green) expression in Hepa1-6 cells are shown. Nuclei were stained with DAPI. Scale bar, 20 μm. (D) mRNA expression levels of mTOR, STAT3, c-Raf, MET, ERK1, and ERK2 were measured by qRT-PCR. Data are expressed as mean ± SD (n = 3). *$P$ < 0.05 and ** $P$ < 0.01 were considered statistically significant, as assessed using a t-test.

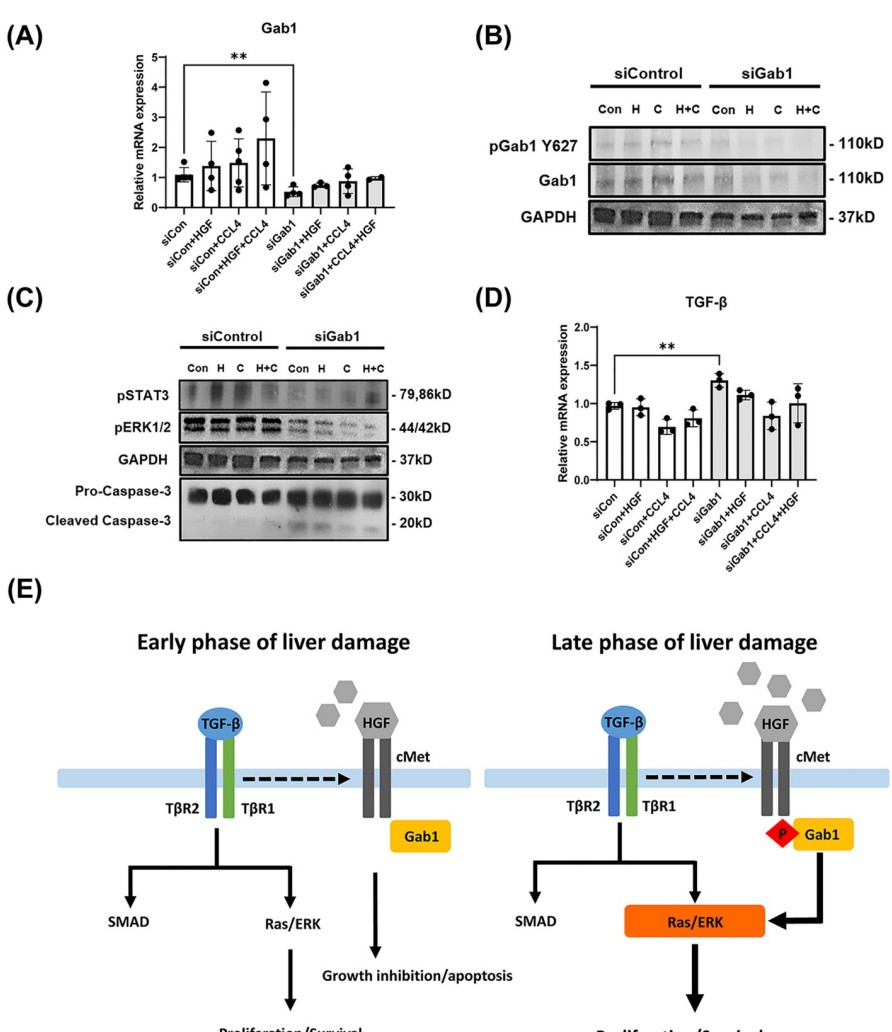

**Fig 6. Gab1 gene silencing reduces cell proliferation and induces cell death signal in mouse hepatoma cell line treated with CCL4 and HGF.** Hepa1-6 cells were treated with carbon tetrachloride (CCL4) and transfected with either AllStars Negative Control siRNA (5 nM) (Qiagen) or Gab1 siRNA (20 pM) (Sigma) using Lipofectamine™ 2000 Transfection Reagent (Thermo Fisher Scientific), following the manufacturer's protocol. Mouse hepatocytes growth factor (mHGF) was added 2 days post transfection for 30 min, and cell lysates were used for analysis. (A) mRNA expression level of Gab1 was measured by qRT-PCR (n = 4). (B) Protein levels of pGab1, Gab1 and GAPDH were assessed by western blot analysis. (C) Protein levels of pSTAT3, pERK1/2, GAPDH, pro-caspase-3, and cleaved caspase-3 were assessed by western blot analysis. (D) mRNA expression of TGF-β is presented (n = 3). (E) Schematic diagram of the possible role of Gab1 in liver disease. During the early phase of liver damage, the production of TGF-β activates TGF-β signaling, leading to cell growth inhibition and apoptosis. Subsequently, damaged hepatocytes activate cell proliferation pathway and develop chronic liver disease such as fibrosis via pGab1 activation of the PI3K/AKT and ERK pathways. ** $P < 0.01$ was considered statistically significant, as assessed using a t-test.

Taken together, these results provide a mechanistic model to explain the difference between early and late host responses during liver disease progression. During the early phase of liver damage, TGF-β production and its signaling to TGF-β receptor in hepatocytes downregulate cMet/pGab1, leading to cell growth inhibition and apoptosis. Ongoing hepatic inflammation at later stage of liver damage can trigger cell survival and proliferation signals, promoting chronic liver diseases such as fibrosis. Moreover, our studies demonstrate that in chronic liver

disease induced by CCL4, PI3K/AKT and ERK pathway activation through phosphorylated Gab1 leads to increased cellular proliferation and survival (Fig 6E).

## Discussion

In the present study, we described the role of Gab1 in the development of liver fibrosis via HGF/c-Met signaling axis. Gab1 is an adaptor molecule that plays a crucial role in the cell survival pathways via binding to various growth receptors such as receptor tyrosine kinase c-Met, Shp2, and Grb2 proteins [22]. Gab1 expression is elevated in various cancers and is involved in carcinogenesis and metastasis [6, 23]. The increased level of Gab1 is strongly correlated with poor prognosis in HCC patients [24]. Moreover, the tyrosine 627 site of Gab1 activates ERK and PI3K through Shp2 binding [22, 25]. Our studies indicate that the levels of TGF-β signaling-related proteins were increased whereas the levels of hepatocyte pGab1(Y627) were decreased in humanized liver mice following acute viral infection. These results suggest a potential role of Gab1 in regulating TGF-β signaling. Interestingly, kinetic studies for the expression of Gab1 in infected hepatoma cells revealed an inverse relationship between Gab1 and TGF-β synthesis. We conducted the in vivo study using the carbon tetrachloride (CCL4)-induced liver fibrosis mouse model to clarify the changes in Gab1 expression in chronic liver disease. CCL4 is a hepatotoxic drug that is metabolized by liver enzymes, leading to the production of toxic radicals, DNA mutations, and abnormal protein/lipid metabolism, which cause extensive liver damage and trigger an inflammatory response [26]. Liver diseases are characterized by increased hepatocyte proliferation, which is associated with upregulation of cMet and/or HGF. HCC promote hepatocyte proliferation and regeneration, as well as tumor angiogenesis through induction of VEGF expression and increased cMet activity [27, 28]. This promotes the development and progression of chronic liver disease such as HCC, leading to rapid tumor growth and poor patient prognosis [29].

Notably, in the liver of fibrotic mice, both Gab1 mRNA and pGab1(Y627) protein levels are significantly increased compared to that in control mice. Gab1 knockdown in mouse hepatoma cells treated with CCL4 alone, or CCL4 plus HGF, showed a significant increase in cleaved caspase-3, along with decreased pERK1/2 and pSTAT3, as compared to control cells. However, the mRNA expression of c-Raf and mTOR was found to be consistent with that of Gab1 expression. This suggests that cell survival-related signaling proteins (e.g., ERK, STAT3, cRaf, and mTOR) play a role downstream of Gab1 signaling. It also indicates that ERK activation is simultaneously affected by both TGF-β and Gab1, but STAT3 and c-Raf expression may depend on Gab1 activation.

Gab1 is an adapter molecule that binds to the receptor tyrosine kinase c-Met upon various extracellular stimuli such as growth factors and cytokines. Gab1 plays a critical role in fibrogenesis and tumorigenesis in various cancers, including HCC [23, 30]. Interaction of Gab1 and c-Met leads to activation of downstream signaling of Ras/Raf and PI3K/AKT signaling axis involved in cell survival and proliferation. Imbalance of Gab1 activity plays an important role in cancer progression and metastasis in a wide range of tumors. Gab1 phosphorylation at tyrosine residues induces rapid hepatocyte proliferation after acetaminophen (APAP)-induced liver damage [5]. Moreover, aggressive tumor progression and poor prognosis in patients with HCC are closely associated with a significant increase in Gab1 expression [22]. Conflicting results for a pathogenic role of Gab1 in liver fibrosis have been reported. Loss of Gab1 aggravates liver fibrosis in the mouse model [3], while Gab1 contributes to the prevention of liver fibrogenesis [31].

However, the relationship between Gab1 expression and the outcome of liver diseases is still unclear. Based on our findings and previous reports, it is likely that cross-regulation of

TGF-β and Gab1 signaling may play a role in the regulation of liver diseases. TGF-β is a pleo-tropic cytokine that is a key regulator of liver disease during various stages of disease progression, from early liver injury to fibrosis and HCC. While TGF-β induces apoptosis of hepatocytes, aberrant TGF-β activity triggers excessive cell proliferation and tumorigenesis. In the early stages of liver injury, TGF-β plays opposing roles in hepatocytes and hepatic stellate cells (HSCs), respectively. TGF-β induces HSC activation, leading to the deposition of fibrous collagen in the extracellular matrix, and promotes the progression of liver disease [7]. However, TGF-β promotes the differentiation of hepatocytes into malignant tumor cells in the late stages of liver diseases such as cirrhosis/HCC [8].

One interesting question raised by present work is how Gab1 is involved in the progression of liver fibrosis. The hepatocyte growth factor (HGF)/c-Met pathway has been reported to play an important role in TGF-β-mediated liver disease progression. Interestingly, although both pathways contribute to cancer cell proliferation and metastasis by upregulation, a recent study revealed that the c-Met pathway is negatively regulated by TGF-β [32]. This finding allows speculation about the role of Gab1, one of the substrates of c-Met, and suggests that a crosstalk between HGF/c-Met and TGF-β signaling may contribute to regulation of the severity of liver disease.

Our findings suggest a role for Gab1 in the development of liver fibrosis. Accordingly, in the early stages of liver disease, the TGF-β signaling pathway in hepatocytes is enhanced while Gab1(Y627) activation is decreased (Fig 6E). In contrast, during chronic liver disease progression, there is an increase in pGab1(Y627) and the expression of molecules involved in cell survival and proliferation pathways, as compared to the early stages of the disease. Conversely, the expression of TGF-β may be relatively weak. Gab1 activates the downstream signaling pathways involved in cell proliferation [33]; it binds to two sites (Y1349, Y1356) of the cytoplasmic tail of c-Met, which closely interacts with numerous signaling molecules such as Gab1, PI3K, PLC, and SHP2 [34]. In particular, activation of the SHP2 binding site of Gab1 (Y627) enhances and sustains ERK1/2 activation [19]. Our results indicate that the effect of Gab1 and TGF-β on the ERK pathway is dependent on the stage of liver disease progression. As the level of Gab1 activation by cMet and TGF-β can exhibit contradictory functions depending on the stage of liver disease progression, more detailed studies on Gab1 expression are required for a clear understanding of its functional characteristics.

## Supporting information

**S1 Fig. HCV core expression in humanized liver mice.** Humanized liver mice were injected in the tail vein with $1 \times 10^4$ and $1 \times 10^5$ FFU/100 μL of JFH-1 (genotype 2a) diluted in PBS (3 mice/group). Liver tissues were harvested 7 days after virus inoculation. Protein levels of HCV core and GAPDH were analyzed by Western blot.
(TIF)

**S1 Data.**
(XLSX)

**S1 Raw images.**
(PDF)

## Acknowledgments

We thank the members of the Hahn lab for providing critical advice for this work

## Author Contributions

**Conceptualization:** Da-eun Nam, Ramin M. Hakami, Young S. Hahn.

**Data curation:** Da-eun Nam, Soo-Jeung Park, Samson Omole, Young S. Hahn.

**Formal analysis:** Da-eun Nam, Soo-Jeung Park, Samson Omole, Eugene Um.

**Funding acquisition:** Ramin M. Hakami, Young S. Hahn.

**Investigation:** Da-eun Nam, Ramin M. Hakami, Young S. Hahn.

**Methodology:** Da-eun Nam, Soo-Jeung Park, Samson Omole.

**Project administration:** Da-eun Nam, Ramin M. Hakami, Young S. Hahn.

**Resources:** Ramin M. Hakami, Young S. Hahn.

**Software:** Da-eun Nam, Soo-Jeung Park, Samson Omole, Eugene Um.

**Supervision:** Ramin M. Hakami, Young S. Hahn.

**Validation:** Da-eun Nam, Soo-Jeung Park, Eugene Um, Ramin M. Hakami, Young S. Hahn.

**Visualization:** Da-eun Nam, Soo-Jeung Park, Samson Omole.

**Writing – original draft:** Da-eun Nam, Soo-Jeung Park.

**Writing – review & editing:** Da-eun Nam, Soo-Jeung Park, Ramin M. Hakami, Young S. Hahn.

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
