## [Decision Letter · Decision Letter 0]

26 Apr 2024

PONE-D-24-04415Activated Gab1 drives hepatocyte proliferation and anti-apoptosis in liver fibrosis via potential involvement of the HGF/c-Met signaling axisPLOS ONE

Dear Dr. Park,

Thank you for submitting your manuscript to PLOS ONE. After careful consideration, we feel that it has merit but does not fully meet PLOS ONE’s publication criteria as it currently stands. Therefore, we invite you to submit a revised version of the manuscript that addresses the points raised during the review process.

We look forward to receiving your revised manuscript.

Kind regards,

Mahmoud Kandeel

Academic Editor

PLOS ONE

Journal Requirements:

2. To comply with PLOS ONE submissions requirements, in your Methods section, please provide additional information regarding the experiments involving animals and ensure you have included details on (a) methods of sacrifice, (b) methods of anesthesia and/or analgesia, and (c) efforts to alleviate suffering.

- https://doi.org/10.1002/hep.32042

(among others)

In your revision ensure you cite all your sources (including your own works), and quote or rephrase any duplicated text outside the methods section. Further consideration is dependent on these concerns being addressed.

a.

- Y.S.H.

- Grant R01 DK122737

- National Institutes of Health

b. 

- R.M.H

- Grant R42 AI122666-03

- National Institutes of Health

5. We note that your Data Availability Statement is currently as follows: All relevant data are within the manuscript and its Supporting Information files.

7. PLOS ONE now requires that authors provide the original uncropped and unadjusted images underlying all blot or gel results reported in a submission’s figures or Supporting Information files. This policy and the journal’s other requirements for blot/gel reporting and figure preparation are described in detail at https://journals.plos.org/plosone/s/figures#loc-blot-and-gel-reporting-requirements and https://journals.plos.org/plosone/s/figures#loc-preparing-figures-from-image-files. When you submit your revised manuscript, please ensure that your figures adhere fully to these guidelines and provide the original underlying images for all blot or gel data reported in your submission. See the following link for instructions on providing the original image data: https://journals.plos.org/plosone/s/figures#loc-original-images-for-blots-and-gels. 

Reviewers' comments:

Reviewer's Responses to Questions

**Comments to the Author**

1. Is the manuscript technically sound, and do the data support the conclusions?

Reviewer #1: Yes

Reviewer #2: Partly

2. Has the statistical analysis been performed appropriately and rigorously? 

Reviewer #1: Yes

Reviewer #2: I Don't Know

3. Have the authors made all data underlying the findings in their manuscript fully available?

Reviewer #1: Yes

Reviewer #2: Yes

4. Is the manuscript presented in an intelligible fashion and written in standard English?

Reviewer #1: Yes

Reviewer #2: Yes

5. Review Comments to the Author

**Reviewer #1:** The authors have presented a well written and rigorously conducted experiment. Figures and tables have been provided to demonstrate the results and statistical analyses conducted. The conclusion obtained from the study provides important insight into the role of Gab-1 in the therapeutics of chronic liver disease.

**Reviewer #2**: PLoS ONE – PONE-D-24-04415 – Park et al.

The authors investigated the role of Gab1, an adaptor protein for various growth factor and cytokine receptors (including HGF, a hepatocyte growth factor), that is involved with cell differentiation and survival signaling pathways in the progression of liver fibrosis. Increased level of Gab1 is strongly correlated with poor prognosis in patients with hepatocellular carcinoma. Cellular and molecular biology methodology was used. Using hepatocytes isolated from humanized mice after acute viral infection, the authors studied signaling molecules related to the progression of liver diseases. A decreased level of pGab1 was detected simultaneously with an increased activation of TGF-β- related cell survival signaling pathways; TGF-β-mRNA expression was higher in infected cells compared to control. Because of the suggestion that apoptosis and proliferation signals were simultaneously activated in hepatocytes following viral infection, the authors hypothesized that Gab1 may participate in regulating the severity of liver disease progression by interacting with the TGF-β signaling pathway. Kinetic studies indicated that Gab1 expression was inversely related to the production of TGF-β during viral infection in hepatocytes. However, when using an established murine model of liver fibrosis induced by carbon tetrachloride treatment, Gab1 mRNA and relative protein levels increased in comparison to the control. Gab1 was thus suggested to play a role in progression of chronic liver disease by regulating cell proliferation. Using Gab1 knockdown cells, the results suggested that the expression of Gab1 is induced by HGF-c-Met signaling axis and that it is involved in increasing cell proliferation/survival and TGF-β expression. The authors concluded that there are differences between early and late host response during liver disease progression. In the early stages of liver disease, the TGF-β signaling pathway is enhanced while Gab1 activation is decreased; during chronic liver disease pGab1 and the expression of molecules involved in cell survival/proliferation pathways increase.

The subject of this manuscript is very interesting and the methodology was judiciously chosen to solve complementary aspects of the results. Certainly, further studies are required to fully understand the contradictory results on Gab1-mediated signaling pathway.

Materials and Methods – It was not mentioned how picrosirius-stained regions were measured. Equipment? Methodology?

Results – Lines 292-293 – Apparently, it cannot be assumed that mRNA expression of ERK has increased. No statistical significance was indicated in Figure 2. Similarly, increase in cleaved caspase-3 cannot be attributed because p = 0.31 is not accepted to mean statistical significance (Line 297).

Line 327 and Figure 3A – Apparently, a significant increase of Gab1 mRNA occurred until D1. Line 330 and Figure 3D – The same consideration applies to ERK.

Fig. 4 – E does not refer to mRNA expression. Figure elements should be described sequentially in the text. Description of D should precede E or restructure of the position of the figure elements should be undertaken.

Fig. 5 – There is something wrong with (C); the column named Merge shows only DAPI staining. Legend: There is no ***P< 0.001 in this figure

Fig. 6 – (E) Insert “Liver damage” over the line where “Early phase” and “Late phase” appear written. Legend: *P< 0.05 and ***P< 0.001 do not refer to this figure.

Minor:

. Standardize: min or minutes

. Lines 128, 130, 133, 134 … 3 mm, 10 min, 30 min, 100 µm, … respectively

. Line 137 – Insert “assay” after blue

. Lines 146, 147, 243 – The first time the commercial source is mentioned, please add city and country

. Line 154 – “mm2”

. Lines 160, 375,395,402 – typos

. Line 171 – “described [17]”

. Line 178 – a verb is missing

. Lines 246-247 – Here, (Sigma, Aldrich) is sufficient. City and country should be added to Sigma mention at line 147.

References – 1. Please abbreviate the name of the journal

6. PLOS authors have the option to publish the peer review history of their article (what does this mean?). If published, this will include your full peer review and any attached files.

Reviewer #1: No

Reviewer #2: No

---

## [Author Response · Author response to Decision Letter 0]

6 Jun 2024

We appreciate for reviewers’ efforts and consideration of the manuscript for publication. Our responses to reviewers’ comments are described below.

Response to Reviewer 1’ comments

The authors have presented a well written and rigorously conducted experiment. Figures and tables have been provided to demonstrate the results and statistical analyses conducted. The conclusion obtained from the study provides important insight into the role of Gab-1 in the therapeutics of chronic liver disease.

Response: We appreciate for the reviewer’s positive comments on the manuscript.

Response to Reviewer 2’ comments

Comment 1

Materials and Methods – It was not mentioned how picrosirius-stained regions were measured. Equipment? Methodology?

.

Response: We added the contents related to picrosirus red staining in Methods section (line 194-199)

Comment 2

Results – Lines 292-293 – Apparently, it cannot be assumed that mRNA expression of ERK has increased. No statistical significance was indicated in Figure 2. Similarly, increase in cleaved caspase-3 cannot be attributed because p = 0.31 is not accepted to mean statistical significance (Line 297).

Response: We added the sentence related to “There’s no statistical difference in mRNA expression of ERK” at line 296.

Line 327 and Figure 3A – Apparently, a significant increase of Gab1 mRNA occurred until D1. 

Line 330 and Figure 3D – The same consideration applies to ERK.

Response: “D2” was revised as “D1” in line 331.

Fig. 4 – E does not refer to mRNA expression. Figure elements should be described sequentially in the text. Description of D should precede E or restructure of the position of the figure elements should be undertaken.

Response: The order of the Fig 4D, E, and F has been changed for sequential explanation (Fig4 and line 369-372).

Fig. 5 – There is something wrong with (C); the column named Merge shows only DAPI staining. Legend: There is no ***P< 0.001 in this figure

Response: We revised “Merge” as “DAPI”, and deleted “***P< 0.001” in the Fig.5 legend. (line 431-432)

Fig. 6 – (E) Insert “Liver damage” over the line where “Early phase” and “Late phase” appear written. Legend: *P< 0.05 and ***P< 0.001 do not refer to this figure.

Response: We inserted “Liver damage” into the Fig6E. and we also deleted “*P< 0.05” and “ ***P< 0.001” in the Fig6 Legend. (line 447)

Comment 3

Minor:

. Standardize: min or minutes

Response: “minutes” was revised as “min” in line 99 and 155.

. Lines 128, 130, 133, 134 … 3 mm, 10 min, 30 min, 100 µm, … respectively

Response: All spacing correction has been completed. (line 128, 130, 133, 134)

. Line 137 – Insert “assay” after blue

Response: We added the “assay” after blue in line 137.

. Lines 146, 147, 243 – The first time the commercial source is mentioned, please add city and country

Response: We added the city and country the first time a commercial source was mentioned. (line 145-146)

. Line 154 – “mm2”

Response: We changed the “2” as superscript in line 155.

. Lines 160, 375,395,402 – typos

Response: We didn’t find mistake on spelling in line 160, 375, 395, 402.

. Line 171 – “described [17]”

Response: We added a space between “described” and “[18]” (ref 17 was changed as 18) in line 171.

. Line 178 – a verb is missing

Response: We revised the sentence including a verb like “There were 150 antibodies used in the Reverse-phase protein microarray data study (Table 1).” in line 177-178.

. Lines 246-247 – Here, (Sigma, Aldrich) is sufficient. City and country should be added to Sigma mention at line 147.

Response: We revised it as “Sigma-Aldrich” in line 250.

References – 1. Please abbreviate the name of the journal

Response: The name of the journal was corrected as abbreviation.

---

## [Decision Letter · Decision Letter 1]

16 Jun 2024

Activated Gab1 drives hepatocyte proliferation and anti-apoptosis in liver fibrosis via potential involvement of the HGF/c-Met signaling axis

PONE-D-24-04415R1

Dear Dr. Park,

We’re pleased to inform you that your manuscript has been judged scientifically suitable for publication and will be formally accepted for publication once it meets all outstanding technical requirements.

Kind regards,

Mahmoud Kandeel

Academic Editor

PLOS ONE

Additional Editor Comments (optional):

Reviewers' comments:

Reviewer's Responses to Questions

**Comments to the Author**

1. If the authors have adequately addressed your comments raised in a previous round of review and you feel that this manuscript is now acceptable for publication, you may indicate that here to bypass the “Comments to the Author” section, enter your conflict of interest statement in the “Confidential to Editor” section, and submit your "Accept" recommendation.

Reviewer #1: All comments have been addressed

Reviewer #2: All comments have been addressed

2. Is the manuscript technically sound, and do the data support the conclusions?

Reviewer #1: Yes

Reviewer #2: Yes

3. Has the statistical analysis been performed appropriately and rigorously? 

Reviewer #1: Yes

Reviewer #2: I Don't Know

4. Have the authors made all data underlying the findings in their manuscript fully available?

Reviewer #1: Yes

Reviewer #2: Yes

5. Is the manuscript presented in an intelligible fashion and written in standard English?

Reviewer #1: Yes

Reviewer #2: Yes

6. Review Comments to the Author

Reviewer #1: (No Response)

Reviewer #2: With your revision, I have no pending questions. Please verify that the typos you did not identify moved to other lines in the revised text:

line 160 - "descried"

line 379 - "Hydorxylproline"

lines 399 and 406 - "hepa"

7. PLOS authors have the option to publish the peer review history of their article (what does this mean?). If published, this will include your full peer review and any attached files.

Reviewer #1: No

Reviewer #2: No

---

## [Editor Report · Acceptance letter]

19 Jun 2024

PONE-D-24-04415R1 

PLOS ONE

Dear Dr. Park, 

I'm pleased to inform you that your manuscript has been deemed suitable for publication in PLOS ONE. Congratulations! Your manuscript is now being handed over to our production team.

Kind regards, 

on behalf of

Professor Mahmoud Kandeel 

Academic Editor

PLOS ONE